Association between visceral fat and bone mineral density in perimenopausal women

Tang Xu 1 2
Tang Ling 2
Li Xiaolin 1
Cao Jiejing 1
Wang Huanhuan 2
Liu Shujiao 2
Yi Yufang 2 13978308157@163.com
Zhang Zhiyong 1 3 4 rpazz@163.com
1 Department of Environmental and Occupational Health, Guangxi Medical University , Nanning, Guangxi , China
2 Department of General Medicine, Affiliated Hospital of Guilin Medical University , Guilin, Guangxi , China
3 Department of Public Health, Guilin Medical University , Guilin, Guangxi , China
4 Guangxi Health Commission Key Laboratory of Entire Lifecycle Health and Care , Guilin, Guangxi , China
Guan Fanglin
Electronic publication date: 2025 Feb 13
Publication date: 2025
Volume: 13
Electronic Location ID: e18957
Received 2024 Oct 24; Accepted 2025 Jan 17
Copyright: © 2025 Tang et al.
Copyright year: 2025
Copyright holder: Tang et al.
License: This is an open access article distributed under the terms of the Creative Commons Attribution License, which permits unrestricted use, distribution, reproduction and adaptation in any medium and for any purpose provided that it is properly attributed. For attribution, the original author(s), title, publication source (PeerJ) and either DOI or URL of the article must be cited.
License URL: https://creativecommons.org/licenses/by/4.0/

Keywords: Visceral fat, Perimenopausal, Bone mineral density, Vitamin D, Osteoporosis

Funding: National Natural Science Foundation of China 81960583 Major Science and Technology Projects in Guangxi AA22096026 Guangxi Science and Technology Development Project AD 17129003 and 18050005 Guangxi Medical and health key-cultivation discipline construction project Guangxi Natural Science Foundation for Innovation Research Team 2019GXNSFGA245002 Guilin science and technology project 20210227-7-11 Guangxi Scholarship Fund of Guangxi Education Department of China The study was supported by the National Natural Science Foundation of China (Grant No. 81960583), Major Science and Technology Projects in Guangxi (AA22096026), the Guangxi Science and Technology Development Project (Grant Nos. AD 17129003 and 18050005), Guangxi Medical and health key-cultivation discipline construction project, the Guangxi Natural Science Foundation for Innovation Research Team (2019GXNSFGA245002), Guilin science and technology project (20210227-7-11) and the Guangxi Scholarship Fund of Guangxi Education Department of China. The funders had no role in study design, data collection and analysis, decision to publish, or preparation of the manuscript.

==============================
Background

The effects of visceral fat and body fat on osteoporosis (OP) have long been controversial. This study investigated the correlation between visceral fat and bone mineral density (BMD) in perimenopausal women aged 40–60. The goal was to evaluate the current state of BMD and its influencing factors, with the specific objective of establishing a foundation for preventing and treating osteoporosis in this demographic.

Methods

This case-control study included female participants (n = 330), aged 40–60 years, from the Health Management Center of Guilin Medical College Affiliated Hospital, China, between January 2020 to August 2023. Their BMD was assessed using an ultrasound bone mineral density meter, and the visceral fat area was determined utilizing a body composition analyzer. Furthermore, past medical history, dietary habits, and lifestyle factors were collected through a telephonic questionnaire survey. Additionally, we analyzed the baseline characteristics of the population, bone status and visceral fat status, and the relationship between these variables.

Results

Among perimenopausal women with varying bone mineral density statuses, there was no significant difference regarding body fat percentage (p = 0.359). In contrast, a statistically significant difference was observed regarding visceral fat area (p < 0.001) and vitamin D (p < 0.001). The visceral fat area exhibited an inverse correlation with bone density (r = –0.313, p < 0.001). Additionally, mediation analysis outcomes did not support the hypothesis that visceral fat affects bone density through its influence on vitamin D levels (p = 0.92).

Conclusions

Among perimenopausal women, visceral fat is negatively associated with bone density, suggesting that the distribution of body fat rather than the total amount plays a pivotal role in the development of osteoporosis. These findings suggest the significance of regular physical exercise and the abdominal fat distribution for perimenopausal women.

Introduction

The perimenopausal period, defined as the time from its onset until a year after the last menstrual period, marks a significant transition for women from adulthood to later life. Due to factors associated with declining ovarian function, women in the perimenopausal phase are at high risk of developing osteoporosis (Seifert-Klauss et al., 2012).

Previous research has associated higher body mass index (BMI) with greater bone density, attributed to the increased mechanical stress on bones associated with weight gain, thus promoting bone tissue growth and maintenance (Xiao et al., 2022). Research indicates that an elevated BMI may be associated with a reduced risk of osteoporosis, particularly in the lumbar spine and hip regions (Zhang, Tan & Tan, 2023). Additionally, a higher BMI generally signifies better nutritional status, a protective factor against osteoporosis. BMI tends to reflect the overall fat content of the body, while visceral fat more accurately describes fat distribution. The two complement each other. Usually, a higher BMI in populations correlates with increased levels of body and visceral fat, although the relationship is not entirely linear. With the continuous advancement in fat measurement techniques, more accurate data regarding fat content and distribution become attainable. While visceral fat has been linked to the progression of cardiovascular disease and diabetes (Chen et al., 2022; Jia et al., 2020), the relationship between with bone density remains unclear. The association between visceral fat and bone density has long been controversial. Some researchers argue that the body’s four major components—fat, water, bone mass, and lean body mass—contribute to bone density, with increased fat content being advantageous (Geoffroy et al., 2019; Kim et al., 2022). Fat tissue can impact bone density through mechanical stress, and fat cells can also secrete estrogen, leptin, inflammatory factors, and other substances that promote bone density.

Conversely, it has been reported that even after eliminating confounding factors such as body weight, there exists a negative correlation between fat tissue content and bone density (Mele et al., 2022). The mechanisms underlying the effect of fat distribution on bone density are complex. Studies suggest that fat distribution, particularly visceral and hip fat, may substantially impact changes in bone density among older men. In contrast, visceral fat is an essential negative regulator of bone density among middle-aged and older women. During the perimenopausal phase, women experience changes in estrogen levels, leading to increased body fat content and decreased muscle mass. Consequently, this process contributes to central obesity, characterized by the accumulation of visceral fat (Maimoun et al., 2016; Starup-Linde et al., 2022).

Given the above background, A case-control study was conducted to explore the correlation between lifestyle factors and bone density in perimenopausal women. Furthermore, we aim to investigate the association between visceral fat and bone density and its potential underlying mechanisms. To explore whether visceral fat can influence bone density by affecting vitamin D metabolism.

Methods

Study subjects

This case-control study recruited female participants aged 40–60 years, from the Health Management Center of Guilin Medical College Affiliated Hospital, China, between January 2020 and August 2023. The sample size required for one-way ANOVA analysis was calculated using PASS 2021 with the following parameters: power = 0.9, alpha = 0.05, number of groups = 3, group allocation pattern = 2:2:1. Based on preliminary survey data, the mean visceral fat for the three groups were 100, 105, and 110, with a standard deviation of 17. The final calculated sample size required was 265. The study subjects were included based on predetermined inclusion and exclusion criteria. The inclusion and exclusion criteria were set as follows:

Inclusion criteria: ① Female residents aged 40–60 years and lived in Guilin for a minimum of 6 months.

② Female experiences menstrual irregularities and remains in this state within 1 year after reaching menopause.

③ Females who have reached menopause but were less than 1 year old.

④ Participants with complete information and signed the informed consent form.

Exclusion criteria: ① Volunteers undergoing osteoporosis treatment, continuous treatment with antipsychotic, antidepressant, or statin medications.

② Potentially pregnant women.

③ Volunteers with conditions such as hyperthyroidism, hyperparathyroidism, pituitary disorders, fractures, diabetes, cancer, rheumatoid arthritis, or other metabolic diseases.

④ Women who have used contraceptive or hormonal medications in the past 6 months or have undergone premature ovarian failure, hysterectomy, or oophorectomy.

⑤ Volunteers with skin conditions preventing exposure to sunlight.

⑥ Women who have recently used medications or received intravenous injections affecting bone metabolism.

Finally, a total of 330 complete samples were collected, and all of them signed the informed consent form (Fig. 1). Participants underwent various assessments, including ultrasound bone density evaluation, body composition analysis, and measurement of vitamin D (VD) levels. Participants were categorized into different groups based on their T-scores: the osteoporosis group (N = 18), those with a T-score ≤ –2.5; the control group (N = 231), those with a T-score ≥ –1.0; and the osteopenia group (N = 81), those with a T-score between –2.5 and –1.0.

Figure 1 A flowchart of patient inclusion.

Questionnaire

The Women’s Osteoporosis Health Factors Questionnaire was used to include common osteoporosis risk factors, The questionnaire has been uploaded as an attachment. This questionnaire was developed by the author of this article. The questionnaire was subjected to repeated measures testing. The questionnaire includes basic information, menstrual history, past medical history related to bone mineral density, and lifestyle habits that may influence osteoporosis. These items included in the questionnaire were determined through extensive literature review and evaluation by clinical experts (Fatima, Brennan-Olsen & Duque, 2019; Hill & Aspray, 2017; Malmir, Larijani & Esmaillzadeh, 2020; Min, Yoo & Choi, 2021). All questionnaires were completed through telephone follow-up, and the investigators underwent standardized training.

Blood and physical indicators

In the fasting state, venous blood samples were collected from the participants at the Health Management Center of Guilin Medical College Affiliated Hospital for biochemical analysis. The tests included serum 25-hydroxyvitamin D, complete blood count, liver function tests, renal function tests, lipid profile, blood glucose, and other components, using a blood biochemical analyzer (Cobas 8000, Roche Ltd., Basel, Switzerland).

The body composition analysis of the study subjects, including body fat percentage and visceral fat area, was conducted using the In Body S10 model body composition analyzer (InBody370, InBody Co., Ltd., Cheonan-si, Korea).

The SONOST3000 ultrasound bone densitometer (SONOST3000, Osteosys Co., Ltd, Seoul, Korea) was utilized to assess ultrasound bone density. The calcaneus is the measurement site.

Statistical analyses

Statistical analyses were conducted using R version 4.1.3. Data conforming to a normal distribution underwent one-way analysis of variance, while non-normally distributed data were analyzed using the Kruskal-Wallis test. The least significant difference method was employed for post-hoc comparisons in one-way analysis of variance. Logistic regression analysis was employed to identify potential influencing factors. Variables with differences between groups in Table 1 were included in the same logistic regression model. Since age is a recognized influencing factor for osteoporosis, we have also included it as a covariate (Shieh et al., 2022; Yang, Wang & Cong, 2022). The dependent variable was defined as the control group and the abnormal bone mass group, wherein the latter group comprised the combination of the osteoporosis group and the osteopenia group. In mediation analysis, we adjusted for age and BMI as covariates. Partial correlation analysis, controlling for age and BMI, was used to determine the relationship between visceral fat area and bone density. Causal mediation analysis was used to examine the mediating effects of vitamin D. A significance level of α = 0.05 was maintained throughout the analyses.

Table 1 Comparison of various indexes among female participants.

	Control (N = 231)	Osteopenia (N = 81)	Osteoporosis (N = 18)	F/H/χ2	p	
Age (years)	49.10 ± 5.12	48.81 ± 5.28	49.61 ± 4.83	0.198	0.821	
Weight (kg)	56.64 ± 4.20	56.83 ± 4.98	57.42 ± 3.31	0.294	0.743	
BMI (kg/cm2)	23.07 ± 2.00	23.17 ± 1.90	24.19 ± 1.11	2.798	0.062	
Visceral fat area (cm2)	99.02 ± 10.65	108.21 ± 13.80	120.72 ± 14.73	41.429	<0.001*	
Body fat percentage (%)	33.57 ± 2.83	33.97 ± 3.68	34.42 ± 2.17	1.027	0.359	
Plasma VD level (ng/mL)	18.90 (12.2)	17.23 (12.82)	9.50 (12.44)	17.657	<0.001*	
Milk intake				24.106	<0.001*	
	No	46 (19.91%)	33 (40.74%)	11 (61.11%)			
	Yes	185 (80.09%)	48 (59.26%)	7 (38.89%)			
VD intake				4.144	0.126	
	No	191 (82.68%)	65 (80.25%)	18 (100.00%)			
	Yes	40 (17.32%)	16 (19.75%)	0 (0.00%)			
Calcium intake				11.389	0.003*	
	No	105 (45.45%)	28 (34.57%)	14 (77.78%)			
	Yes	126 (54.55%)	53 (65.43%)	4 (22.22%)			
Daily sunlight exposure time (min)				14.905	0.001*	
	<30	112 (48.72%)	60 (73.20%)	11 (61.11%)			
	≥30	118 (51.28%)	22 (26.80%)	7 (38.89%)			
Age at menarche (years old)				5.648	0.227	
	≤12	33 (14.29%)	17 (20.99%)	6 (33.33%)			
	13~16	177 (76.62%)	58 (71.60%)	11 (61.11%)			
	≥17	21 (9.09%)	6 (7.41%)	1 (5.56%)			
Mode of delivery				2.256	0.324	
	Eutocia	153 (66.23%)	47 (58.02%)	13 (72.22%)			
	Caesarean	78 (33.77%)	34 (41.98%)	5 (27.78%)			
Notes:

* p < 0.05.

BMI, body mass index; VD, vitamin D.

Ethics approval and consent to participate

All subjects gave their informed consent for inclusion before they participated in the study. The study was conducted in accordance with the Declaration of Helsinki, and was reviewed and approved by the ethics committee of Affiliated Hospital of Guilin Medical University before the investigation. Thesis Proposal of the ethical approval document (No. 2019GLMUIAY069), dated August 22, 2019. Ethical Approval (No. 2022YJSLL-94) expiry date of August 25, 2023.

Results

Baseline characteristics of the study subjects

As presented in Table 1, significant differences were observed among different groups regarding visceral fat area, plasma vitamin D concentration, milk intake, calcium intake, and daily sunlight exposure time (p < 0.05). However, there were no statistically significant differences among the groups regarding age, weight, BMI, body fat percentage, vitamin D intake, age at menarche, and mode of delivery (p > 0.05).

As shown in Fig. 2, post-hoc comparisons obtained from one-way analysis of variance revealed statistically significant differences in visceral fat among all groups. Additionally, statistically significant differences in vitamin D levels were observed between the osteopenia group and the control group, as well as between the osteoporosis group and the control group.

Figure 2 Distribution of biomarkers among different experimental groups.

Note: Group comparisons were performed using one-way ANOVA followed by S-N-K for subsequent comparisons.**p < 0.01; ns p > 0.05.

Factors of osteoporosis

As shown in Table 2, age, visceral fat area, plasma vitamin D concentration, milk intake, calcium intake, and daily sunlight exposure time were all identified as risk factors for abnormal bone mass occurrence. The occurrence rate of the abnormal bone mass showed a positive correlation with age (p = 0.002, odds ratio (OR) = 1.108) and visceral fat area (p < 0.001, OR = 1.068) and exhibited a negative correlation with plasma vitamin D concentration (p = 0.006, OR = 0.951). The occurrence rate of abnormal bone mass was 2.955 times higher in individuals who did not consume milk compared to those who did (p < 0.001, OR = 2.955), and was 2.146 times higher in individuals who did not take calcium supplements compared to those who did (p = 0.026, OR = 2.146), and was 1.918 times higher in individuals with less than 30 min of daily sunlight exposure compared to those with more than 30 min (p = 0.034, OR = 1.918).

Table 2 Logistic regression analysis of abnormal bone mass.

	β	Stand error	Wald	p	OR	95% CI	
Age (years)	0.102	0.034	9.192	0.002*	1.108	[1.037–1.183]	
Plasma VD level (ng/mL)	–0.05	0.018	7.555	0.006*	0.951	[0.918–0.986]	
Visceral fat area (cm2)	0.065	0.012	27.45	<0.001*	1.068	[1.042–1.094]	
Milk intake							
	No	1.083	0.311	12.147	<0.001*	2.955	[1.607–5.434]	
	Yes				–	–	–	
Calcium intake							
	No	0.764	0.343	4.963	0.026*	2.146	[1.096–4.202]	
	Yes				–	–	–	
Daily sunlight exposure time (min)							
	<30	0.652	0.308	4.476	0.034*	1.918	[1.049–3.508]	
	≥30				–	–	–	
Notes:

* p < 0.05.

CI, confidence interval; OR, odds ratio.

Correlation between visceral fat area, vitamin D, and BMD

As shown in Table 3, when controlling for age and BMI in partial correlation analysis, we found a negative correlation between visceral fat area and bone density in perimenopausal women (r = –0.313, p < 0.001). Additionally, there was a positive correlation between vitamin D levels and BMD (r = 0.288, p < 0.001).

Table 3 Correlation between visceral fat area, vitamin D, and BMD.

	BMD	
	r	p	
Visceral fat area (cm2)	–0.313	<0.001	
Plasma VD level (ng/mL)	0.288	<0.001	
Note:

BMD, bone mineral density.

Mediation analysis

Mediation analysis was conducted with plasma vitamin D concentration as the mediator, visceral fat area as the independent variable, bone density as the dependent variable (Table 4). We observed that plasma vitamin D concentration did not significantly mediate the relationship between visceral fat area and bone density (p = 0.92, Estimate = 0.036). The visceral fat area exerted direct impact in bone density (p < 0.001, Estimate = –4.314).

Table 4 Analysis of the mediation effect of visceral fat area on bone density through vitamin D.

	p	Estimate	95% CI	
Average causal mediation effects	0.08	0.003	[–0.006 to 0.001]	
Average direct effects	<0.001*	–0.032	[–0.038 to –0.020]	
Total effect	<0.001*	–0.034	[–0.042 to –0.020]	
Proportion mediated	0.08	0.069	[–0.009 to 0.210]	
Note:

* p < 0.05.

Discussion

Factors of osteoporosis

This study revealed that among perimenopausal women, advanced age and elevated visceral fat area are risk factors for abnormal bone mass. Higher plasma vitamin D levels, regular milk consumption, calcium supplementation, and daily sunlight exposure of more than 30 min have been identified as protective factors against osteoporosis. The protective effects associated with higher plasma vitamin D levels, milk consumption, calcium supplementation, and daily sunlight exposure exceeding 30 min offer valuable insights into potential preventive strategies against osteoporosis. Adequate vitamin D levels are crucial for calcium absorption and bone mineralization, highlighting the importance of maintaining optimal vitamin D status through dietary intake and sunlight exposure. Furthermore, milk and calcium supplements provide additional dietary sources of calcium, which is essential for maintaining bone density and strength. Moreover, regular sunlight exposure exceeding 30 min can promote endogenous vitamin D synthesis, further supporting bone health.

BMI and BMD

This study found no significant difference in BMI (which can be substituted by weight and body fat percentage) between the abnormal bone mass group and the control group, which is inconsistent with some previous studies (Jia & Cheng, 2022; Liu et al., 2021; Tang et al., 2023). Jia & Cheng’s (2022) study did not provide baseline data, making it impossible to assess the overall BMI level. Liu et al. (2021) and Tang et al.’s (2023) studies indicated that only excessively low BMI contributes to increased bone mass loss, and in populations with normal BMI or overweight, no significant correlation was found between BMI and BMD. Some studies even suggest that overweight and obesity may affect bone metabolism, leading to bone mass reduction. The latest overweight criterion for Asians is a BMI greater than 23, and the average BMI of the participants in this study was above 23, so strictly speaking, the results do not completely conflict with other studies. BMI does not directly influence bone mineral density. A very low BMI often indicates poor nutritional status, leading to increased bone mass loss (Gkastaris et al., 2020; Wang et al., 2023; Zhao, Xu & Leung, 2022). Research conducted by Choi, Hong & Lim (2013) has indicated that insulin resistance in obese individuals leads to elevated levels of insulin, proinsulin, and preptin hormones, which can significantly disrupt bone cell metabolism. For individuals with normal or even overweight BMI, an increase in BMI does not result in a significant gain in BMD.

Visceral fat and osteoporosis

This study found that visceral fat area is negatively correlated with bone BMD in perimenopausal women, and excessive visceral fat is a risk factor for the occurrence of osteoporosis. This is consistent with the findings of Sun et al. (2023), who conducted a study using NHANES data. It is worth noting that individuals with higher BMI often exhibit higher body fat content (Auslander et al., 2022), which has led some researchers to inappropriately perceive a high body fat ratio as a protective factor against osteoporosis (Gkastaris et al., 2020). Studies have indicated that as women age, there is a gradual reduction in the percentage of muscle in the body, accompanied by an increase in fat ratio. This transition, particularly evident in lean tissue among perimenopausal women aged 50–59, significantly impacts bone density, while fat ratio remains relatively unchanged (Akash et al., 2018; Boutens et al., 2018). For women over 60 years in the postmenopausal stage, body fat composition becomes a crucial factor influencing bone density, as fat tends to accumulate in different areas such as the abdomen, viscera, and upper arms, leading to central obesity. Usually, the distribution of body fat is influenced by endocrine and hormonal levels. Even among individuals with similar body fat ratios, significant variations in fat distribution may exist, particularly concerning abdominal fat, which is associated with various diseases.

Research conducted in the United States has revealed a link between visceral fat and BMD among adolescent girls, where visceral fat emerged as a detrimental factor for BMD, corroborating previous findings (Russell et al., 2010). Molecular-level investigations have suggested that this relationship may stem from the secretion of various immune and inflammatory factors by visceral fat, as well as various pathways affecting lipid and bone metabolism, along with the role of oxidative stress in inhibiting bone turnover markers and reducing bone mineralization (Crivelli et al., 2021; Hilton et al., 2022; Li et al., 2023). Furthermore, research indicated that an increase in abdominal fat can diminish the secretion of adiponectin while enhancing the release of tumor necrosis factor-alpha (TNF-α). Adiponectin promotes the secretion of osteocalcin, type I collagen, and alkaline phosphatase, facilitating the development of osteoblasts. In contrast, TNF-α activates RANKL to promote osteoclastogenesis, making bones more susceptible to resorption (Berner et al., 2004).

Moreover, the presence of visceral fat area influences bone density among perimenopausal women, providing scientific evidence for the prevention and treatment of osteoporosis (Crivelli et al., 2021; Hilton et al., 2022; Li et al., 2023). The association between visceral fat and osteoporosis has received considerable attention in academic literature. Previous studies have consistently demonstrated a negative correlation between visceral fat accumulation and BMD, particularly in postmenopausal women (Li et al., 2023, 2020). Visceral fat is metabolically active and can influence systemic conditions such as insulin resistance, hyperglycemia, and dyslipidemia, and all of which adversely affect bone health. Moreover, visceral fat deposition is associated with hormone changes, including decreased estrogen levels in postmenopausal women, exacerbating bone loss (Rył et al., 2020; Sun et al., 2023). Collectively, these findings suggest a complex and multifactorial relationship between visceral fat accumulation and osteoporosis. We highlighted the need for further investigations to elucidate underlying mechanisms and develop targeted interventions to mitigate the adverse skeletal effects of visceral adiposity (Yang et al., 2022; Zhu et al., 2020). The relationship between visceral fat and bone mineral density is complex and multifactorial. While vitamin D metabolism and mechanical loading are important pathways, other mechanisms such as oxidative stress and adipocyte-derived factors like adipokines and inflammatory cytokines may also influence bone health (Chávez Díaz et al., 2019). Further studies are needed to clarify how these factors interact with visceral fat to impact bone density.

Vitamin D

Furthermore, this study identified a positive correlation between plasma vitamin D levels and bone density, aligning with previous research findings (Anagnostis et al., 2020; Aspray et al., 2019). Combining the negative correlation between visceral fat area and bone density, we performed further analysis to explore whether vitamin D acts as a mediator in this relationship. However, the results indicate that vitamin D does not serve as a mediator between visceral fat area and bone density, suggesting that visceral fat does not influence bone density through calcium metabolism pathways. Considering osteoporosis as a multifactorial disease, supplementation with vitamin D and calcium alone may offer limited preventive effects against osteoporosis (Burt et al., 2019; Hill et al., 2019).

Suggestion

The results of this study suggest that along with traditional osteoporosis prevention strategies, perimenopausal women should engage in physical exercise and pay attention to their abdominal fat storage, especially the rational distribution of body fat components. Based on our findings, the following behaviors are recommended: (1) Exercise interventions: engaging in moderate-intensity aerobic exercise and resistance training. (2) Dietary interventions: controlling total energy intake, reducing saturated and trans-fat intake, and increasing intake of vegetables, fruits, whole grains, and low-fat dairy products. (3) Other interventions: ensuring sufficient sleep, controlling weight, smoking and alcohol consumption, and undergoing regular BMD screening. This approach targets multiple pathways to inhibit the decline in bone density, effectively preventing the onset of osteoporosis. If needed, people should go to health management centers to develop prevention strategies that suit them.

Limitation

There are some limitations in our study. Firstly, The case-control design of this study, while useful for exploring associations, limits the generalizability of the findings to broader populations. Differences in lifestyle, genetic background, and environmental factors in other populations may lead to variations in the observed relationships. Secondly, additional information, such as lifestyle, gravidity, parity of the participants, bone turnover markers, sex hormone levels, and economic, psychological, and environmental factors, was not collected (Gao et al., 2023), which will be included in our future studies. Anyway, this article provides some new clues for the prevention and treatment of osteoporosis.

Conclusions

Among perimenopausal women, visceral fat is negatively associated with bone density, indicating that the distribution of body fat rather than the total amount plays a crucial role in the development of osteoporosis. These findings suggest the significance of regular physical exercise and the abdominal fat distribution for perimenopausal women.

Supplemental Information

Supplemental Information 1 Raw data.

Supplemental Information 2 Codebook.

Supplemental Information 3 STROBE checklist.

The authors are very grateful to the participants who kindly contributed to this study. We thank the staff of the Clinical Laboratory, Affiliated Hospital of Guilin Medical University.

Abbreviations

BMI body mass index

BMD bone mineral density

CI confidence interval

OR odds ratio

OP osteoporosis

VD Vitamin D

Additional Information and Declarations

Competing Interests

The authors declare that they have no competing interests.

Author Contributions

Xu Tang conceived and designed the experiments, prepared figures and/or tables, authored or reviewed drafts of the article, and approved the final draft.

Ling Tang analyzed the data, prepared figures and/or tables, and approved the final draft.

Xiaolin Li conceived and designed the experiments, prepared figures and/or tables, and approved the final draft.

Jiejing Cao analyzed the data, authored or reviewed drafts of the article, and approved the final draft.

Huanhuan Wang analyzed the data, authored or reviewed drafts of the article, and approved the final draft.

Shujiao Liu performed the experiments, authored or reviewed drafts of the article, and approved the final draft.

Yufang Yi analyzed the data, authored or reviewed drafts of the article, and approved the final draft.

Zhiyong Zhang performed the experiments, authored or reviewed drafts of the article, and approved the final draft.

Human Ethics

The following information was supplied relating to ethical approvals (i.e., approving body and any reference numbers):

The study was conducted in accordance with the Declaration of Helsinki, and was reviewed and approved by the ethics committee of Affiliated Hospital of Guilin Medical University before the investigation. Thesis Proposal of the ethical approval document (No. 2019GLMUIAY069), dated August 22, 2019. Ethical Approval (No. 2022YJSLL-94) expiry date of August 25, 2023.

Data Availability

The following information was supplied regarding data availability:

The raw measurements are available in the Supplemental Files.

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
