# Peer review of "Association between visceral fat and bone mineral density in perimenopausal women"

_PeerJ, doi:10.7717/peerj.18957_

## Round 0.1 · original submission · Major Revisions

Based on the comments provided by both reviewers, significant revisions are required before further consideration. Please carefully address all points raised, particularly the methodology and data interpretation. A detailed point-by-point response to the reviewers' comments should be provided with your revised manuscript.

Reviewer 1 ·

Basic reporting

The research design and methodology of the manuscript are appropriate, and the findings are of some value, providing clinical evidence for the ongoing debate. However, the conclusions of the study are limited due to the lack of support from basic biological research and the fact that the study sample only includes individuals from a specific region. The language of the manuscript needs to be more precise, as many details have been overlooked, and it may only be accepted after revisions.Comments on this manuscript are as follows:

1.The introduction highlights the importance of osteoporosis in perimenopausal women but lacks sufficient discussion on the molecular mechanisms linking visceral fat to bone density. Adding a more detailed explanation of how visceral fat impacts bone metabolism would strengthen the theoretical framework.
2.All the disputes about the effect of visceral fat on bone density mentioned in this paper are the results of epidemiological studies. Can it be considered that the most lack of this phenomenon is basic research on biological mechanisms, which also reflects the lack of innovation in this study?
3.Bone mineral density is detected by means of ultrasound, rather than the accepted dual-energy X-ray detection. The test position is not mentioned in the method section. The bone mineral density of different parts varies greatly among individuals. Please be sure to supplement it.
4.The number of samples in the low bone mass group in Table 1 and the inconsistency in Figure 1, please carefully check and confirm.
5.The title in the article does not mention vitamin D. Why did the author include vitamin D as a mediator variable in the model and spend a lot of space to describe the relationship between vitamin D and each variable? Do you need to modify the title?
6.The study recruited 330 subjects. How is this sample size being determined? If justified, please state it in the method.
7.The conclusion provides general lifestyle suggestions but lacks specificity. Including more actionable, evidence-based recommendations tailored for healthcare providers or patients would increase the paper's practical value.
8.BMD in Table 1 is classified in three groups, while in Table 2 there are two groups. Please explain why this is done and whether this has an impact on the final result.
9.The introduction in the article mentioned that the effect of fat on bone density may be bidirectional, but the relationship between visceral fat is chosen to test the relationship with bone density. Why not use a restrictive cubic spline to detect possible nonlinear relationships.
10.The manuscript suggests an association between visceral fat and BMD, but it would benefit from a more comprehensive discussion on potential biological pathways linking these two variables, such as inflammation or hormonal alterations. This could strengthen the paper's scientific contribution.

Experimental design

No more

Validity of the findings

No more

Reviewer 2 ·

Basic reporting

This study investigated the association between visceral fat and bone mineral density (BMD) in perimenopausal women. It found that visceral fat, not overall body fat, was negatively correlated with BMD. This suggests that targeted lifestyle interventions focused on reducing abdominal fat could help prevent osteoporosis. While the study explores an important topic, it has several limitations that affect the validity and generalizability of its findings:
1. While the introduction covers relevant background information, the flow could be improved by better connecting the different points. For example, the transition between the paragraphs discussing BMI and visceral fat could be smoother.
2. The concluding sentence of the introduction, while stating the study's aim, could be strengthened by providing a more specific and concise statement of the research question or hypothesis.
3. 330 subjects were included, so how this number was determined.
4. The results in Table 1 mentions significant differences in visceral fat area, vitamin D levels, milk intake, calcium intake, and sunlight exposure time across the groups. These factors are known to influence bone health and may confound the association between visceral fat and bone density. The study needs to consider these potential confounders in its analysis to determine if the observed relationship between visceral fat and bone density is truly independent of these other factors.
5. The results in Table 2 doesn't explicitly mention if the logistic regression was adjusted for other factors potentially impacting bone density. This is crucial to determine whether the identified factors independently contribute to abnormal bone mass or if their effects are mediated by other variables.
6. How does the mediation model mentioned in Table 4 handle the covariates? If specific adjustment models are lacking, the conclusions will become unreliable.
7. The discussion presents various findings and their implications but doesn't clearly articulate the study's main conclusions or contributions to the field. It would be beneficial to state the study's key findings in a concise and impactful way, emphasizing their significance.
8. While the discussion mentions some mechanisms, it's important to acknowledge the complexity of the relationship between visceral fat and bone density. Exploring potential alternative explanations, such as oxidative stress or adipocyte-derived factors, would demonstrate a deeper understanding of the topic and strengthen the discussion. In addition, the discussion should delve deeper into the potential impact of the limitations. For example, it could discuss how the case-control design might affect the generalizability of the findings or how the lack of information on specific factors could have influenced the results.
9. In some places, the meaning could be clearer. For example, the sentence "Consequently, a research suggests that higher BMI may reduce the risk of osteoporosis..." is a bit confusing. It might be better to say, "Some research suggests that higher BMI may reduce the risk of osteoporosis..."
10. While the sentences are generally grammatically correct, some sentences are quite long and complex. Breaking them down into shorter, more concise sentences would enhance readability. There are a few instances where articles (a, an, the) are used incorrectly, such as "a research suggests" instead of "some research suggests." Pay attention to the correct use of articles.

Experimental design

/

Validity of the findings

/

Additional comments

/

---

## Round 0.2 · Major Revisions

While the two reviewers have recommended acceptance, upon careful review by myself, several important issues still need to be addressed as follows,

1. The statistical approach for variable selection in the logistic regression analysis needs refinement. Rather than relying solely on previous studies to include variables like age, we recommend using a more systematic approach (e.g., including variables with p < 0.1 in univariate analysis).
2. There is an inconsistency between your extensive discussion of BMI's relationship with osteoporosis in the introduction and its exclusion from the logistic analysis. Please justify this decision or revise accordingly.
3. The sample size calculation using PASS software requires clarification. Please explain how you arrived at the figure of 265 participants and provide the parameters used in this calculation.
4. Please verify the sample size (n=231) reported for the Low bone mass group in Table 1.
5. The study design (prospective vs. retrospective) needs clear specification, as there appears to be inconsistency between the abstract and methodology sections.
6. Please revise "Objectives" to "Background" for better alignment with standard manuscript structure.
7. The description of study subjects needs significant clarification. Please provide:
- Clear numbers for both control and experimental groups before and after screening
- Specific inclusion/exclusion criteria for both groups
- A flow diagram showing participant selection and attrition
8. For the questionnaire developed by the author, please provide:
- The methodology used for question selection
- Validation process (if any)
- Rationale for included items
9. All materials and instruments should be described in the format: Material/Instrument Name (Catalog Number, Company, City, Country).
10. Please provide a high-resolution version of Figure 1.
11. The results shown in Figure 1 warrant more detailed description, particularly regarding BMI comparisons. Consider expanding this section to highlight key findings that may not be apparent in Table 1.
12. The reported lack of significant differences in weight, BMI, and body fat percentage between groups contradicts existing literature. Please discuss this discrepancy and its implications for your findings.

Please address these concerns comprehensively in your revision.

Reviewer 1 ·

Basic reporting

no comment

Experimental design

no comment

Validity of the findings

no comment

Reviewer 2 ·

Basic reporting

The manuscript has been refined and the current version basically meets the requirements for publication.

Experimental design

no comment

Validity of the findings

no comment

---

## Round 0.3 · accepted · Accept

I have thoroughly reviewed your detailed responses and revisions to the manuscript. All the comments have been adequately addressed with clear explanations and appropriate modifications, including the statistical methodology, study design clarification, sample selection process, and enhanced discussion of findings. The manuscript has been significantly improved, and can be accepted for publication.